# Synthesis of Retinol-Loaded Lipid Nanocarrier via Vacuum Emulsification to Improve Topical Skin Delivery

**DOI:** 10.3390/polym13050826

**Published:** 2021-03-08

**Authors:** Seung-Hyun Jun, Hanul Kim, HyeJin Lee, Ji Eun Song, Sun Gyoo Park, Nea-Gyu Kang

**Affiliations:** LG Household and Health Care R&D Center, Seoul 07795, Korea; hanul59@lghnh.com (H.K.); hellohj1223@lghnh.com (H.L.); sos6934@lghnh.com (J.E.S.); skparke@lghnh.com (S.G.P.)

**Keywords:** retinol encapsulation, nanostructured lipid nanocarrier, vacuum emulsification, thermal stability, penetration efficiency, low inflammatory factors

## Abstract

Retinol has been widely used as an anti-wrinkle active ingredient in cosmetic fields. However, the oxidation of retinol by air was one of the critical problems for application in the skincare field. In this study, Retinol-loaded lipid nanocarriers were prepared via the vacuum emulsification method to increase the stability of retinol vulnerable to air and optimized encapsulation conditions and to increase the penetration efficiency into skin. Optimizing the components of lipid nanocarriers, gradients of carbon chain C8-22 using various lipid species which made the amorphous structure and enough spaces to load retinol inside the capsules were estimated from the lower enthalpy change and peak shift in DSC analysis. The vacuum-assisted lipid nanocarriers (VLN) could help suppress oxidation, which could have advantages to increase the thermal stability of retinol. The retinol-loaded VLN (VLN-ROL) had narrow size distribution under 0.3 PDI value, under 200 nm scaled particle size, and fully negative surface charge of about -50 mV for the electrostatic repulsion to avoid aggregation phenomenon among the lipid nanoparticles. It maintained 90% or more retinol concentration after 4 weeks of storage at 25, 40 and 50 °C and kept stable. The VLN-ROL-containing cream showed improved penetration efficiency applied to porcine skins compared to the commercial retinol 10S from BASF. The total amount of retinol into the skin of VLN-ROL (0.1% of retinol) was enhanced by about 2.2-fold (2.86 ± 0.23 μg) higher than that in 0.1% of bare retinol (about 1.29 ± 0.09 μg). In addition, applied on a 3D Human skin model, the epidermal thickness and the relative percentage of dermal collagen area effectively increased compared to the control and retinol, respectively. Additionally, the level of secreted IL-1α was lower and epidermal damage was weaker than commercial product A. This retinol-loaded lipid nanocarrier could be a potentially superior material for cosmetics and biomedical research.

## 1. Introduction

Skin aging is defined as all the changes in the skin envelope that result from the temporal accumulation of gradual changes in its various constituent components. Skin aging can be classified in two categories: a) photo-aging caused by ultraviolet (UV) light and b) natural aging caused by temporal accumulation. The most representative changes caused by aging are skin wrinkles. Skin wrinkles cause a flat surface of epithelial and dermal–epithelial joints, while in the case of the dermis, the production of collagen and elastin fibers by fibroblasts is gradually decreased by certain enzymes, and by an overall decrease in the extracellular substrate (ECM) manifested by an increase in the destruction of these polymers [1].

Several specific compounds have already been identified as anti-wrinkle activators and have been used specifically as cosmetic products for the purpose of preventing signs of skin aging, reducing and/or removing skin wrinkles. Among them, vitamin A, also commonly referred to as retinoids (retinoic acid, retinol, retinal, and retinoid derivatives) has also proven to reduce wrinkles in cosmetics and medicines by improving the thickness of the epidermis and by strengthening the ECM layer. Accordingly, special medications (retinoic acid) and cosmetics (retinol and retinoid derivatives) have been extensively used to improve wrinkles [2,3,4]. In spite of its several benefits, however, some drawbacks in the use of retinoid should be overcome owing to its side effects, low solubility, and rapid degradation. Retinoids can cause side effects, such as pruritus, burning sensation, erythema, peeling, and conjunctivitis around the eyes. Additionally, retinoids are easily photodegradable by exposure to heat, air, or light, so their efficacies decrease over time; furthermore, there is concern about stimulation by photodegradation substances [3,5]. Specifically, all-trans-retinol, which can be used as an active ingredient in the field of cosmetics, can be easily oxidized to carboxylic acid in the alcohol group of retinol, and makes retinol susceptible to degradation. For example, pure retinol degrades within 2 days at 4 °C and at a pH of 7.0 [6], thus indicating that pure retinol is extremely unstable and challenging to formulate for cosmetic and pharmaceutical applications. Therefore, various methods used for stabilizing retinoids have been reported. For example, retinyl palmitate, hydroxypinacolone retinoate (HPR) and retinyl retinoate, which combine the carboxyl group at the end of the retinoid with other substances, have been reported and are being released as a product [7,8,9]. The potency of the retinoids is strongly dependent on its metabolic distance to retinoic acid. These retinol derivatives was metabolized into retinol, and then retinol was oxidized into retinal, which subsequently be oxidized to retinoic acid as metabolism of retinyl esters to retinoic acid [10]. Therefore, the retinoid-like activity of retinol derivatives after topical application is lower than that of retinol. Another approach has been studied for the stabilization of retinol. Specifically, nanodispersed systems, such as liposomes [11], nanoemulsions [12], polymeric encapsulation [13], electrostatic interaction [14] and lipid nanoparticles [15,16], have gained increasing importance as potential vehicles for the controlled delivery of cosmetics and for the optimized disposition of active ingredients in particular skin layers. Among these methods, lipid nanoparticles, such as solid lipid nanoparticles and nanostructured lipid carriers (NLC) for retinol encapsulation, have been studied extensively owing to well-tolerated carrier systems for dermal applications [15,16]. Recently, lipid nanoparticles made from a complex lipid mixture called smartLipids^®^ was introduced to achieve increased loading, and a firmer inclusion inside the particle matrix was more protective for chemically labile molecules [17].

Numerous techniques have been developed for the production of lipid nanoparticles. These methods include high-pressure homogenization [18,19,20], emulsification-solvent evaporation [21], emulsification-solvent diffusion [22,23], solvent injection [24], phase inversion [25,26], and multiple emulsion [27]. Among them, the high-pressure homogenization technique has many advantages compared with other methods, such as easy to scale up, avoidance of organic solvents, and short production times. Although high-pressure homogenization is a very powerful method for synthesizing lipid nanoparticles, these methods required two processes. Moreover, encapsulation of retinol into lipid nanoparticles could be one of effective means to protect them against chemical degradation, it does not completely remove air contained during the synthesis process. Therefore, other alternatives should be required for practical application of retinol. Herein, we introduce a new approach using a vacuum emulsification method for the synthesis of retinol-based lipid nanocarriers. Vacuum emulsifiers are extensively used in the mixing process produced in the pharmaceutical industry, such as ointments, preparations, gels, and suspensions. The emulsification tank uses a fully enclosed vacuum system. The vacuum emulsifier utilizes the many important vacuum functions for the improvement of the performance of mixing compared with other types of mixers and product handling to achieve the best mixing results. This also allows the material to be homogenized, emulsified, heated, and dispersed in a vacuum environment to reduce the contact of the material with the outside air. Oxidation has been associated with a number of problems. Therefore, vacuum emulsification, which could help reduce oxidation, could offer considerable benefits to increase the stability of active substances vulnerable to air. Specifically, the stability of retinol was rapidly decreased by air, and blocking the air during the synthesis of retinol encapsulation via vacuum emulsification could have an important role in the improvement of the stability of the retinol.

In this study, we prepared retinol-loaded lipid nanoparticles with the vacuum emulsification method by optimizing the synthetic conditions, such as the lipid composition, surfactant type, zeta potential, and pH. The superiority of the retinol-loaded vacuum-assisted lipid nanocarriers (VLN-ROLs) was confirmed based on their improved thermal stability, particle size, polydispersity characteristics, and skin penetration efficiency effects. We also investigated whether these efficacy and side effects occurred in the metabolic pathway of retinoid when three-dimensional skin was used. To measure the effect factors, such as epidermal thickness, dermal collagen, and inflammatory factors, a skin modeling system was established that could simultaneously confirm the efficacy and side effects of retinol-loaded VLNs. It is inferred that VLN-ROLs can maintain a high-retinol content and ensure stability. These cannot only constitute a viable alternative for industrial production, but they are also potentially suitable for use in skincare and biomedical research applications.

## 2. Materials and Methods

### 2.1. Materials

The main components of solid lipids, glyceryl behenate, glyceryl stearate, glyceryl distearate, cetyl palmitate, myristyl myristate, poloxamer 188, Retinol 10S and Retinol 50C were purchased from BASF (AG, Ludwigshafen, Germany). Moreover, caprylic/capric triglyceride was purchased from KLK OLEO (Mutiara Damansara, 47,810 Petaling Jaya, Selangor, Malaysia). Butylated hydroxytoluene (BHT), which has antioxidant properties, was purchased from Jinyang Chemical Co. Ltd. (Busan, South Korea). In addition, polysorbate 60, sodium stearoyl glutamate, and hydrogenated lecithin were obtained from Croda Inc. (Edison, NJ, USA), Ajinomoto Health & Nutrition North America Inc., and from Neuropid (Hwaseong, South Korea), respectively.

### 2.2. Synthesis of Retinol-Loaded Lipid Nanoparticles

This lipid complex-based nanocarrier was synthesized via the vacuum emulsification method. At first, the solid lipids glyceryl behenate, glyceryl stearate, glyceryl distearate, cetyl palmitate, and myristyl myristate were melted at 75 °C with hydrogenated lecithin to achieve an efficient solubilization during mild stirring. The caprylic/capric triglyceride, BHT, and retinol 50 °C were heated at 50 °C during magnetic stirring until perfectly mixed. The three types of surfactants, Tween-60, poloxamer 188, and sodium stearoyl glutamate, were added to the distilled water. The solution was heated at temperatures >80 °C and stirred with magnetic stirring at ~500 revolutions per minute (rpm). When both lipid mixtures were clearly melted, the liquid lipid mixture containing retinol was added to the solid lipid mixture. Consequently, the hot aqueous solution, including surfactants and additives, and the lipid solution, were poured in the vacuum homogenizer and emulsified for 5 min with vigorous stirring. The final solution, which was a dispersion of lipid complex nanoparticles should be cooled at 25 °C and stored at 4 °C until later use. Compositions of lipid complex-based nanocarrier for optimization of synthesis condition were listed in Table 1.

### 2.3. Properties of Retinol-Loaded Lipid Nanoparticles

#### 2.3.1. Stability Study of Retinol in Retinol-Loaded Lipid Nanoparticles

The concentration of retinol in retinol-loaded lipid nanoparticles was analyzed by high-performance liquid chromatography (HPLC) (Nexera ultra high-performance liquid chromatography, Shimadzu, Japan). It consisted of a DGU-405 degassing unit, a LC-40D XR binary pump and a SPD-M40 photodiode array detector. At first, pure retinol powder (Sigma–Aldrich, St. Louis, MO, USA) was used to make standard solutions. It was serially diluted in isopropyl alcohol and ethanol mixture (volume ratio: 50:50) as the solutions have concentrations which are 50, 100, and 300 parts per million. The NLC solutions were also serially diluted 200 times by using same solvent with the standard solution (isopropyl alcohol and ethanol mixture). The sample solutions were ultrasonicated for 5 min to dissolve the retinol inside the particles and filtered with 0.45 um pore-sized filters. The HPLC column was the YMC J’sphere ODS-H80, C18 150 mm × 4.6 mm with an internal particle size diameter of 5 um (YMC, Milford, MA, USA). The column temperature was set to 25 °C and the injection volume was 1 μL. The mobile phase consisted of methanol and water (volume ratio: 90:10). The flow rate was 1 uL/min and the total run time was 14 min, including the washing and equilibration stages. The retention time for retinol was 9.4 min. The UV absorbance of retinol was detected at 325 nm. To investigate the thermal stability of retinol-loaded lipid nanoparticles, all samples were kept in cryo tubes at 25 °C and 40 °C. They were compared with the initial solution which was stored at 4 °C.

#### 2.3.2. Particle Size and Zeta Potential Analysis of Retinol-Loaded Lipid Nanoparticles

The mean diameter of particles and polydispersity index (PDI) in water were measured by dynamic light scattering analysis which uses a Zetasizer Nano-ZS90 (Malvern Instruments Ltd., Malvern, UK), at a fixed angle of 90° and at 25 °C. The NLC solutions were diluted 1500 times in distilled water for size analyses, and 150 times for zeta potential analyses. The measured data on Z-average (nm) and PDI were evaluated based on the intensities of the particles, and were assessed in triplicate.

#### 2.3.3. Differential Scanning Calorimetric (DSC) Analysis of Retinol-Loaded Lipid Nanoparticles

To investigate the information about the phase transition behaviors of retinol-loaded lipid nanoparticles, such as denaturation and melting, DSC analysis was performed with the Perkin-Elmer DSC 4000 analyzer (Perkin Elmer, Wellesley, MA, USA). Approximately 10 mg of the sample were loaded on the closed-type aluminum sample pan and heated from 30 to 80 °C at a scan rate of 10 °C/min. The same empty pan was used as a reference. The temperatures of the melted peaks and enthalpies of the tested samples were calculated by the Pyris manager software, data analysis program supplied by Perkin-Elmer (Perkin Elmer, Wellesley, MA, USA).

### 2.4. Skin Penetration Analysis

To analyze the penetration efficiency of VLN-ROL, porcine skins (APURES Co., Ltd., Pyeongtaek, South Korea) were used. They were isolated from the back skin of pigs, and their thicknesses were 1000 um. Skin penetration experiment using Franz diffusion cell has performed as modified previous study [28]. The porcine skins were mounted on the Franz diffusion cell and the reservoir was filled with phosphate buffer saline and ethanol (50% mixture) to dissolve hydrophobic retinol. The concentrations of retinol in the cream formula were 0.1% and 0.3% which made by retinol-loaded lipid nanoparticles (3.3% and 10% of Restinol^TM^) and retinol 10S for comparison. The porcine skins were incubated in a constant temperature and humidity chamber (37 °C, 50%) for 16 h. After 16 h, the unabsorbed retinol cream was wiped by cotton swabs and the corneous layers were stripped three times by a stripping tape. The skin tissues were shredded by the Precellys 24 homogenizer in ethanol for the elution of retinol. This procedure was repeated twice and the average value of penetration was calculated.

### 2.5. Reconstructed 3D Human Skin

The reconstructed 3D human skin model Neoderm^®^-ED was purchased from Tego Science (Seoul, South Korea) and analyzed according to the manufacturer’s instructions. Briefly, bare retinol, VLN-ROL formulations, or commercial product were applied on the top of the 3D human skin model. Commercial product was oil-in-water cream formula and retinol contents of commercial product was 0.1%. The reconstructed 3D human skin model was incubated at 37 °C at 5% CO_2_. After 48 h, the supernatant sample was collected, and interleukin-1α (IL-1α) was analyzed with a human IL-1 alpha/IL-1F1 DuoSet enzyme-linked immunosorbent assay system (DY200, R&D Systems, Minneapolis, MN, USA). The cultured 3D skin model was fixed with 4% paraformaldehyde, embedded in a paraffin block, and stained with Masson’s trichrome staining. After staining, images of sectioned 3D human skin samples were acquired with the EVOS™ FL Auto2 Imaging System (Thermo Fisher Scientific, Waltham, MA, USA) and epidermis thickness and dermal collagen area were measured with Celleste™ Image Analysis Software (Thermo Fisher Scientific, Waltham, MA, USA).

### 2.6. Statistical Analysis

Data are presented as mean values ± SD. Statistical analysis of data was performed using Student’s t-test and analysis of variance (ANOVA). A *p*-value of less than 0.05 was considered significant.

## 3. Results and Discussion

### 3.1. Synthesis of Retinol-Loaded Lipid Nanoparticles via Vacuum Emulsification

Synthesis of lipid nanoparticles using high-pressure homogenization technique has many advantages compared with other methods, e.g., easy scale up, avoidance of organic solvents, and short production times. High-pressure homogenizers are extensively used in many industries including the pharmaceutical industry, e.g., for the production of emulsions for parenteral nutrition. Although the high-pressure homogenization is a very powerful method for the synthesis of lipid nanoparticles, these methods have some drawbacks, including the fact that they require two processes and volume losses during the high-pressure homogenization process.

The synthesis of retinol-loaded lipid nanoparticles via vacuum emulsification could minimize loss because it can be completed as a one-pot synthesized process without the need for additional movement processes; additionally, scale-up is also an easy and simple process, and has advantages because the vacuum emulsifier for large scale is generally used in pharmaceuticals or cosmetic production. Additionally, in the general emulsification process, the stability of retinol, which is vulnerable to air, is reduced after the capsulation because air exists in the reactor, and because air may exist in the capsule during its synthesis. As shown in Figure 1, in the case of normal emulsification, the particle size is uneven and the thermal stability is also low. Due to these disadvantages, the high-pressure homogenizer is used to reduce the particle size.

In the case of the high-pressure homogenization technique, which is a two-step process executed after the pre-emulsion process, the composition conditions for NLC synthesis in the pre-emulsion step, such as the ratio of solid/liquid lipid and lipid/surfactant, have limitations caused by the instability of the lipid–water phase. The constraints of the components associated with the formation of the capsule owing to the instability of the lipid–water phase in the pre-emulsion process constitute one of the disadvantages of the high-pressure homogenization technique. For example, if the amount of surfactant is less than 10% subject to the conditions of formula #1 in Table 1, a phase separation occurred when the emulsions were cooled in our synthesis condition. However, the retinol-loaded lipid nanoparticles via vacuum emulsification could be synthesized without aggregation or separation even when the content of the surfactant was reduced by one-fifth in same condition (Table 2). This result implied that having a minimum amount of air in the vacuum mixture, a compact interaction between the lipid and the surfactant, and not being aggregated again with a small amount of surfactant (Figure 1) [29].

### 3.2. Optimization of Composition of Retinol-Loaded Lipid Nanoparticles

We also investigated the optimized composition of retinol-loaded lipid nanoparticles (as a vehicle of retinol) for the enhancement of the stability of retinol and for obtaining proper physical properties. The lipids used for the preparation of retinol-loaded lipid nanoparticles were selected according to two parameters, namely, (a) the liquid lipid that was able to dissolve the drug, and (b) the solid lipid that was able to form a homogenous mixture. The solid/liquid ratio of the NLC carriers was determined to be 2:1 in that the stability of the capsules could be maintained, while the loading of retinol was maximized. Two lipid phases, consisting of the solid and liquid phases (caprylic/capric triglyceride), were prepared separately to minimize the exposure time during which the retinol was exposed to heat at high temperatures during the overall synthesis. In order to check retinol degradation during the synthesis, the retinol content of the lipid particles after synthesis was analyzed using HPLC method. As a result, we confirmed that the initial amount of retinol was almost keep although this vacuum emulsification method contains exposure to high temperature for a while (see Appendix A).The size of retinol-loaded lipid nanoparticles affected the penetration efficiency of retinol into the skin barriers because the small size of particles achieved close contact with the stratum corneum, and the increasing surface area of particles with decreasing sizes improved the transfer of encapsulated retinol. The PDI is a value that estimates the width of the size distribution. When the particles are in a totally diverse form, the PDI value is limited to unity. Furthermore, it is favorable that the PDI value is <0.3 because it indicates that the size of NLC has a narrow distribution and the particles effectively include retinol [30].

#### 3.2.1. Effect of Lipid Composition

At first, we introduced the various lipid species to generate carbon chain C8–22 gradients by referring to previously reported (third generation) lipid nanoparticles [17]. Combining various lipids makes the less-ordered lipid particles have imperfections in capsules. Therefore, they can contain large amounts of drugs. The different lengths of the carbon chains provide spatial forms in capsules, and polymorphic transitions during storage are suppressed to minimize the aggregation phenomenon [17]. Among the solid lipid components, the largest amount of lipid species is glyceryl behenate that constitutes the rigid backbone of capsules. Glyceryl stearate, glyceryl distearate, and cetyl palmitate were used to generate carbon chain gradients. As listed in Table 3, the sizes of the retinol-loaded lipid nanoparticles were indicative of the NLC compositions. Formula #1 had a spatial scale >1 μm, which was too large to penetrate the skin barriers. The PDI value was also equal to unity. This means that the retinol-loaded lipid nanoparticles were not synthesized uniformly. As shown in Figure 2, formula #1 acquires a reddish color when stored at 40 °C. As this color change of the retinol-loaded lipid nanoparticle solution affected the entire formula, it is important to maintain the white-to-yellow color after long-term storage.

#### 3.2.2. Effect of Surfactant

The other important physical property of retinol-loaded lipid nanoparticles is the zeta potential, which refers to surface charge densities. The average diameter (Z-ave) and size distribution are affected by lipid contents; however, the zeta potential is mainly influenced by surfactants. This is because the surfactants, which have both hydrophilic and lipophilic groups, work between the different phases and determine the surface charges of the particles. The coagulation phenomenon of particles is regulated by the electrostatic repulsion effect for zeta potentials > |30| mV that leads to the necessary distance between neighboring particles to obtain stable physical properties. When the particles are aggregated and separated, the encapsulated retinol cannot be protected sufficiently. Therefore, we established strategies to reduce the zeta potential to −50 mV because the lipid particles usually had a negative surface charge. This negative charge provided the electrostatic repulsion among particles and maintained the original distances of particles that helped avoid the degradation of retinol.

Regulation of the zeta potential of the lipid nanoparticles was achieved by using various types of nonionic surfactants, such as lecithin, poloxamer 188, and anionic surfactants—normally used in the field of cosmetics—such as cetyl phosphate (CP) and sodium stearoyl glutamate (SSG), at the same time [31]. When the first candidate of anionic surfactants was used, namely, cetyl phosphate (with a negative charge of minus one), the mean diameter (Z-average) of formula #3 was approximately 300 nm. Formula #3 showed monodisperse and small-sized capsule properties. However, in Figure 2, formula #3 of cetyl phosphate yielded unstable color changes and loose flowability when stored, even at 25 °C. Flowability means a kind of flow characteristics indicating how much the fluid flows well. When the lipid nanoparticles are not fully stabilized in aqueous solution the lipids could aggregate themselves and the solutions could become thick and viscous [32,33]. The second candidate of anionic surfactant, sodium stearoyl glutamate had two negative charges, the mean diameters of formula #3 yielded sizes <300 nm and a PDI value of ~0.2. Upon observation, formula #3 suppressed the reddish discoloration, and maintained flowability stable after 8 weeks at 40 °C. The zeta potential of formula #3 changed considerably to negative values to −50 mV.

To make the zeta potential more negative, the contents of anionic surfactants were increased (formulas #2, 3, and 4). As we expected, the surface charge decreased to -62 mV gradually as a function of the anionic surfactant content. The sizes of the particles reduced slightly to 260 nm from 290 nm, and the PDI value was still <0.3 which means the systems is still uniform. When the anionic surfactant condition changed, there is no meaningful change in size and size distribution. In Figure 2, formula #3 yields the best suppression in discoloration and aggregation. This formula can achieve the necessary stable physical properties, such as stable (non-changing) color, viscosity, and separations when applied to cream. Nonionic surfactants could yield the steric repulsion effect which induces additional repulsions between particles besides the charge repulsion effect induced by ionic surfactants. When four types of surfactants are used, including SDC, Poloxamer 188, Tween-80, and lecithin, the separation phenomenon could be suppressed, and the size and stability of the retinol-loaded lipid nanoparticles could be maintained stable for >1 year [31].

#### 3.2.3. Effect of pH

As the retinol stability could be affected by pH conditions, we also regulated the pH by 10% NaOH aqueous solution in formula #5 in Table 1. The original pH of formula #5 was about 6.5 and we adjusted pH at 7.5 and 8. In Figure 2, the colors of retinol-loaded lipid nanoparticles show different pH in the sample numbers, 8–10. The original pH 6.5 shows the most stable color changes after storage and good thermal stability in both 25 and 40 °C for 8 weeks.

According to the optimized condition listed above with the vacuum emulsification method, we finally synthesized an optimized VLN-ROL referred to as “Restinol^TM^“, which are non-aggregating, visually unchanging, and thermally stable. As shown in Table 4, the final properties of Restinol^TM^ have a uniform distribution with a narrow PDI distribution range and values <0.3 (with small particle sizes <200 nm) that is advantageous for material transfer as a retinol carrier. Zeta potential was measured to be less than -50 mV. This value was sufficient to generate an electrostatic repulsion effect. When the visual stability in storage was evaluated at 25 °C, there was no aggregation phenomenon. After 4 weeks of storage at 25 °C, 40 °C, and 50 °C, the retinol content was analyzed with HPLC, and it was confirmed that the content was maintained at >90%. Therefore, we decided to check the skin penetration and efficacy with this composition.

### 3.3. DSC Analysis of Retinol-Loaded Lipid Nanoparticles

The DSC graphs in Table 5 of both compositions #1 and #3 showed two alpha and beta modification forms similar with other findings in the NLC literature [17,34]. The composition #3 modified with a carbon chain gradient in solid lipids showed shallower peaks than #1 in the DSC graphs. This induced changes in the glass transition process or crystal ordering in lattice and retinol-lipid interactions which resulted from changes in the composition of lipids. The calculated enthalpy of beta modification in #3 was approximately 2 times less than #1, which means that the lipid particles of composition #3 were highly amorphous and have more defects in the lipid core. This was attributed to the fact that the enthalpy of less-ordered smart lipids with amorphous phases was lower than the enthalpy of bulk lipids. The reasoning for this was owing to the fact that the melting enthalpy needed to overcome the lattice energy when the system was perfectly crystalline compared with bulk lipids [17,34].

The lattice structure in lipids replaced by more defects, and a noncrystal, amorphous phase, leads to an enhanced drug-loading efficiency. The adequate spaces in the lipid core provide retinol entrapment positions [35]. Furthermore, the melting peak at ~67 °C in composition #1 shifted to 62 °C in composition #3, and the peak was broadened. These melting peak changes are attributed to the small size effect that means that the defects in the crystalline lattice and lipid cores increase and the surface area of nanoparticles is high. In addition, the interactions between the stabilizers or surfactants and lipids at the interface decrease the crystallinity of lipid matrix and increase the stability of the retinol-loaded lipid nanoparticle system. Accordingly, these interactions cause the shifting of the melting peak and the broadening phenomenon.

### 3.4. Penetration Study of Retinol-Loaded Lipid Nanoparticles on Porcine Skin

We applied 100 uL of each oil-in-water (*o*/*w*) cream to the porcine skins which contained VLN-ROL or bare retinol from BASF Retinol 10S (final concentrations of 0.075, 0.1, 0.3%). According to Figure 3, the amounts of retinol which penetrated in the skins and reservoirs from retinol creams were analyzed by the HPLC method. In the same concentration of the retinol formula, retinol was permeated when the retinol-loaded lipid nanoparticles remained in the skins three times more than when retinol 10 S was used owing to the interaction between lipid capsules and the extra cellular lipid matrix in the skin. In the previous literature, the compositions of lipid capsules affected the penetration of any molecule because different lipids have different lipid solubility parameter values (SP). Typically, it is useful to estimate these values given that release and penetration of any active materials by its interaction with skin lipids could occur when SP > 10. The main lipids used in this retinol-loaded lipid nanoparticles, glyceryl behenate and caprylic/capric triglyceride have SP values equal to 9.34 and 10.36, respectively. Therefore, this composition leads to a greater penetration of retinol into the skin [36].

Additionally, the NLC is known to cover the skin by forming a lipid film that has the occlusive effect of nanoscaled particles consequently reducing the evaporation of water from the skin. This effect leads to the increase in the hydration of the skin and enhanced penetration of the active molecules as the film on the skin improved the diffusion of the active ingredient to deeper layers through the more hydrated stratum corneum [37]. Recently, Müller et al. reported that submicron (100–1000 nm) sized lipid particles could have high adsorptive properties for penetration into skin without being legally nanoparticles and therefore no addition of “(nano)” to INCI nomenclature is required in cosmetic field [38]. This report supports that VLN-ROL, under 200 nm sized, have high adsorptive properties for penetration into skin with comfortable materials for use in cosmetic field.

### 3.5. Effects of Retinol-Loaded Lipid Nanoparticles on 3D Skin Model

To examine the effects of retinol-loaded lipid nanoparticles on the human skin, a reconstructed 3D human skin model (Neoderm^®^-DE) was employed. First, 0.1% and 0.3% retinol-loaded lipid nanoparticles were applied on the 3D human skin model. The relative percentage of the dermal collagen area increased at the two tested concentrations (Figure 4a), and the histological staining yielded a blue color in both cases (Figure 4c). In addition, the epidermal thickness was also increased at the two concentrations of retinol-loaded lipid nanoparticle treatments (Figure 4b).

Subsequently, we investigated the effects of VLN-ROL and compared these with pure retinol and other commercial products. We also constructed an appropriate cosmetic formulation which contained 0.1% VLN-ROL and bare retinol, respectively, and purchased commercial product A which also contained 0.1% retinol. First, we confirmed that VLN-ROL increased the relative percentage of dermal collagen area (Figure 5a) compared with control and retinol samples, respectively. The effect of the commercial product A on the dermal collagen area was similar to the retinol-loaded lipid nanoparticle effects. Furthermore, VLN-ROL also increased the epidermal thickness compared with the control, retinol, and commercial product A (Figure 5b). Compared with controls (no treatment), formulations containing three different retinol types seem to cause some degrees of epidermal damage (Figure 5d) owing to the character of formulation and increased secretion of inflammatory factors, such as IL-1α. Specifically, the epidermal damage of retinol for the commercial product A was severe, and the top layer of epidermis seemed to disappear. Pro-inflammatory cytokine IL-1α is known to be released by keratinocytes when the skin epidermis is damaged [39]. Clinically, it was confirmed that the secretion of IL-1α caused acute and chronic skin inflammation by regulating T-cell targeting chemokine production in prepsoriatic skin [40]. The epidermal damage induced by adding retinol formulation seemed to be attributed to increased IL-1α secretion. Nevertheless, the level of secreted IL-1α was lower (Figure 5c), and the epidermal damage was also significantly weaker after VLN-ROL treatment compared with bare retinol and commercial product A.

### 3.6. Practical Application and Future Research Perspectives

For practical application of method of vacuum-assisted lipid nanocarriers, we have successfully produced for the scale-up process about 50 kg of VLN-ROL using 300 kg capacity of commercial vacuum emulsifier with microfluidizer. The size of retinol-loaded lipid nanocarriers via vacuum emulsification and normal emulsification was 142.6 ± 5.0 nm and 187.9 ± 6.8 nm, respectively. The PDI value of retinol-loaded lipid nanocarriers via vacuum emulsification and normal emulsification was 0.265 ± 0.004 and 0.295 ± 0.011, respectively. These results support that synthesis of lipid nanocarriers via vacuum emulsification cloud efficient process to reduce the size and distribution in scale-up process. However, further improvement is still needed for increasing production efficiency. For example, in the case of vacuum emulsification, compared to normal emulsification, the contents may back flow to the vacuum pump under vacuum condition. Currently, the contents are filled up to 20% of total volume of reactor to proceed with the vacuum emulsification. Therefore, we will try about further research on optimization in the process of increasing the capacity per one production.

## 4. Conclusions

The VLN-ROLs were successfully synthesized with vacuum emulsification. VLN-ROLs could help suppress oxidation and could increase the thermal stability of retinol. As the components of the lipid nanocarriers were optimized, gradients of the carbon chain C8–22 with various lipid species were established which generated (a) the amorphous structure and (b) adequate spaces to allow the loading of retinol inside the capsules. Our study is significant as it is the first attempt at applying lipid nanoparticle using vacuum emulsification to confirm physical properties, such as particle sizes of under 200 nm with narrow size distribution, fully negative surface charges with values that are approximately equal to -50 mV for the electrostatic repulsion to avoid the aggregation phenomenon among the lipid nanoparticles, and thermal stability. The improving skin penetration efficiency of VLN-ROL showed that the total amount of retinol into the skin of VLN-ROL (0.1% of retinol) was enhanced by about 2.2-fold (2.86 ± 0.23 μg) higher than that in 0.1% of bare retinol (about 1.29 ± 0.09 μg). In vitro artificial human skin models yielded high-effect and low-inflammatory factors attributed to VNN-ROLs in comparison with other preparations. The 50 kg scale of synthesis of VLN-ROL was successfully produced with 142.6 ± 5.0 nm of mean diameter and 0.265 ± 0.004 of PDI value.

Thus, vacuum-assisted lipid nanoparticles have a great potential not only to serve as a potential vehicle of retinol in cosmetics as well as in other biomedical applications, but also to apply the practical use of other ingredients to increase the oxidative stability. physical properties for applying an active ingredient which is easily oxidized.

## Figures and Tables

**Figure 1 polymers-13-00826-f001:**
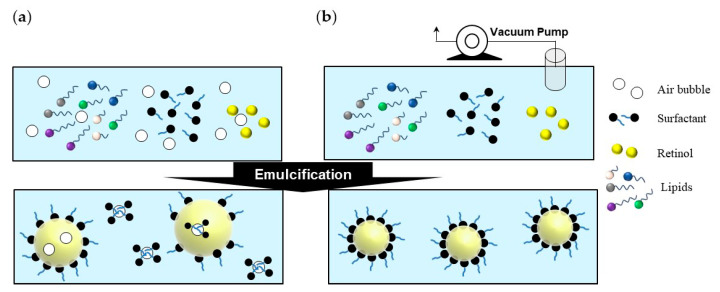
Schematic showing the stepwise differences of the (**a**) normal and (**b**) vacuum emulsification processes.

**Figure 2 polymers-13-00826-f002:**
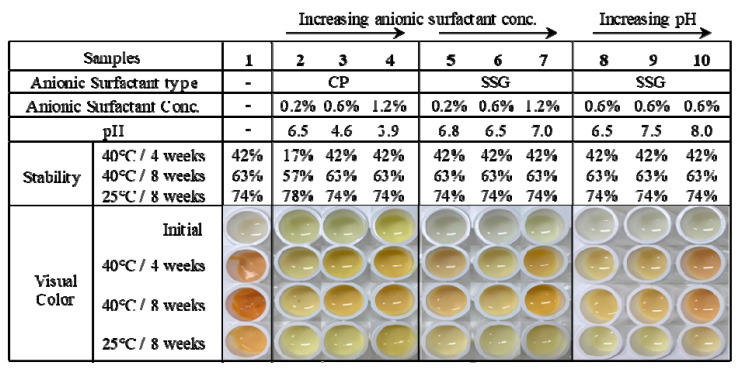
pH values, thermal stabilities, and visual color changes of nanostructured lipid carriers (NLC-ROLs) at different temperatures and durations.

**Figure 3 polymers-13-00826-f003:**
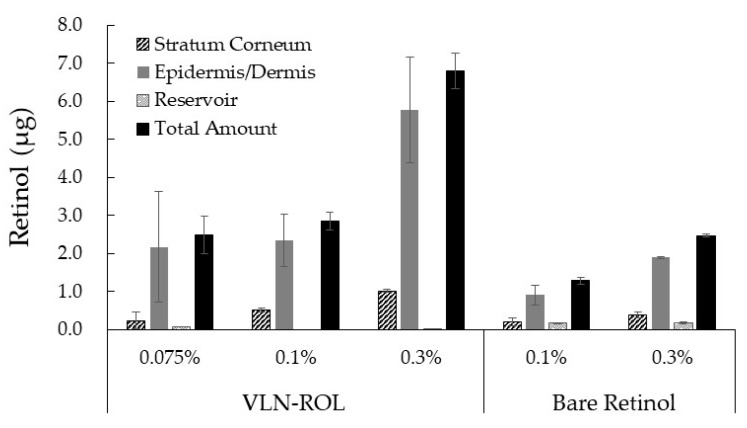
Distribution of retinol permeated in the stratum corneum, epidermis/dermis, and receptor medium, 16 h after the application of o/w cream which contained VCN-ROL and bare retinol at different concentrations of retinol.

**Figure 4 polymers-13-00826-f004:**
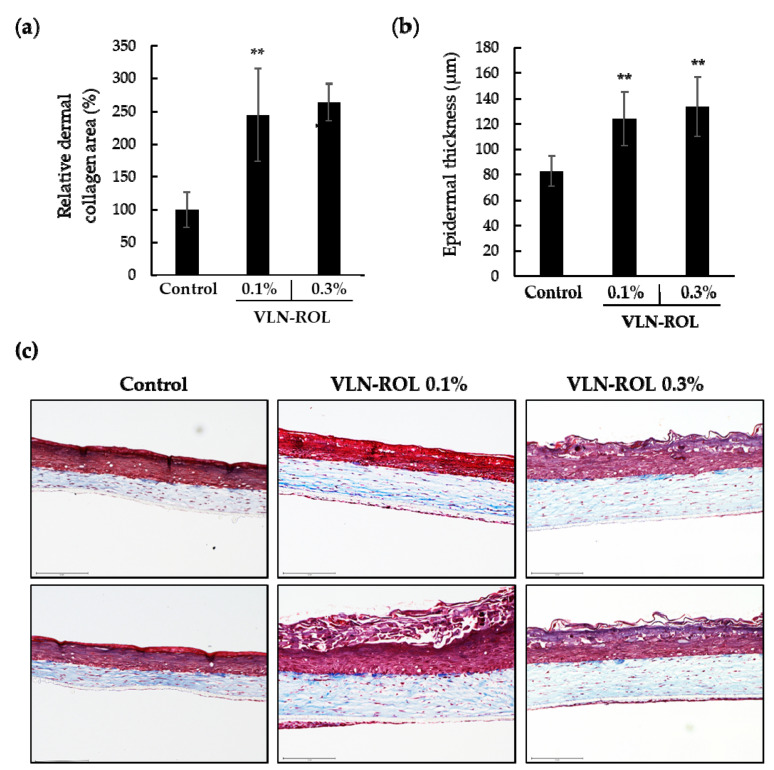
Effects of VLN-ROL on the three-dimensional (3D) human skin model. VLN-ROL was added at different concentration (0.1, 0.3%) on the top of the 3D human skin model and cultured after 48 h. (**a**) Relative dermal collagen area stained by Masson’s trichrome staining and (**b**) epidermal thickness measured with the Celleste™ Image Analysis Software. (**c**) Sectioned 3D human skin images after Masson’s trichrome staining. The upper red color region shows the epidermis area and the lower blue color region shows the dermis area ** *p* < 0.01 indicate significant differences compared with control).

**Figure 5 polymers-13-00826-f005:**
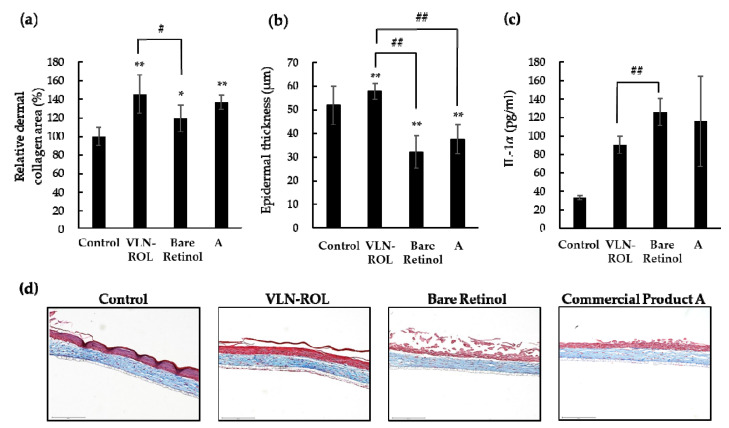
Comparison of effects between VLN-ROL, retinol and commercial product A. Three different formulations containing VLN-ROL and bare retinol, and commercial product A, were, respectively, applied on the top of the 3D human skin model. Each formulation contained 0.1% VNL-ROL or bare retinol. (**a**) Relative dermal collagen area and (**b**) epidermal thickness were both measured with the Celleste™ image analysis software. (**c**) IL-1α secreted by the 3D human skin model was measured. (**d**) Sectioned 3D human skin images after Masson’s trichrome staining (* *p* < 0.05, ** *p* < 0.01, # *p* < 0.05, and ## *p* < 0.01, indicate significantly difference between groups).

**Table 1 polymers-13-00826-t001:** Compositions of retinol-loaded vacuum-assisted lipid nanocarriers (VLN-ROLs).

	Formula	#1	#2	#3	#4	#5
		Weight content				
Solid lipids	Glyceryl behenate	3.6%	3.6%	3.6%	3.6%	3.0%
	Stearylamine	2.4%				
	Glyceryl stearate		0.6%	0.6%	0.6%	0.8%
	Glyceryl distearate		0.6%	0.6%	0.6%	0.8%
	Cetyl palmitate		0.6%	0.6%	0.6%	0.8%
	Myristyl Myristate		0.6%	0.6%	0.6%	0.6%
Liquid lipids	Caprylic/Capric Triglyceride	3.0%	2.4%	2.4%	2.4%	2.4%
	Retinol	3.0%	3.0%	3.0%	3.0%	3.0%
Surfactants	Polysorbate 60	1.8%	0.7%	0.7%	0.7%	0.7%
	Poloxamer 188		0.7%	0.7%	0.7%	0.7%
	Hydrogenated lecithin		0.2%	0.2%	0.2%	0.2%
	Anionic surfactant	0.2%	0.2%	0.6%	1.2%	0.6%
	DI	up to 100				

**Table 2 polymers-13-00826-t002:** Mean diameters (Z-ave) and polydispersity index (PDI) values of retinol-loaded lipid nanoparticles according to the decreasing surfactant concentrations in vacuum emulsification and normal emulsification.

Surfactant Con.(%)	Vacuum Emulsification		Normal Emulsification	
	Z-ave (nm)	PDI	Z-ave (nm)	PDI
10	218.6	0.183	2416	0.593
5	242.0	0.209	Phase separation	
2	405.1	0.172	Phase separation	

**Table 3 polymers-13-00826-t003:** Mean sizes, PDI values, and zeta potentials of nanostructured lipid carriers (NLC)-ROLs containing an anionic surfactant.

Formula	Surfactant	Size (nm)	PDI	Zeta Potential (mV)
#1	-	1386	0.95	12.9
#2	CP	303.5	0.216	−48.7
#3		319.3	0.217	−51.4
#4		306.1	0.196	−54.9
#2	SSG	294.0	0.221	−47.9
#3		269.7	0.201	−54.6
#4		257.9	0.205	−62.2
#5		278.1	0.191	−50.9

**Table 4 polymers-13-00826-t004:** Mean size, PDI value, zeta potential, and stabilities at various temperatures of VLN-ROLs.

Size (nm)	Distribution (PDI)	Zeta Potential (mV)	Thermal Stability (4 Weeks)		
			25 °C	40 °C	50 °C
158.4	0.27	−53.1 ± 3.24	97%	92%	92%

**Table 5 polymers-13-00826-t005:** Melting temperatures and enthalpies of retinol-loaded lipid nanoparticles.

Sample	T_1_(°C)	T_2_(°C)	H_1_ (J/g)	H_2_ (J/g)
1	53.6	67.7	0.23	10.25
3	53.9	61.9	0.82	4.17

## Data Availability

Not applicable.

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
