# Peer review of "Synthesis of Retinol-Loaded Lipid Nanocarrier via Vacuum Emulsification to Improve Topical Skin Delivery"

_polymers, 2021, doi:10.3390/polym13050826_

Round 1
Reviewer 1 Report
I read an interesting and comprehensive research work entitled ‘Synthesis of Retinol-loaded Lipid Nanocarrier via Vacuum Emulsification for Improved Topical Skin Delivery’. The concept of the manuscript completely fits and is suitable to publish in Polymers Journal. This manuscript is generally well written and clearly presented however still needs to address many comments and thus require major revision.
- Title should be modified as ‘Synthesis of Retinol-loaded Lipid Nanocarrier via Vacuum Emulsification to improve Topical Skin Delivery’.
- Abstract section should be rewritten and look very general and not informative. In abstract authors should mention the values of results and importance of research work in one or two sentences. Provide a nice graphical abstract representing the overview of the MS with key highlights.
- In the introduction section, write the novelty of the work and the problem statement clearly. Line no 45-46 how retinol derivatives are useful to improve wrinkles give details.
- Line no 71-77 no reference has been cited which is quiet surprising. Thus detailed discussion about the novelty, significance of your research work and research gap relative to the literature is essential.
- Statistical analysis of the results should be provided in the materials and methods section. It's important for all experimental work Report these values in the results and discussion.
- Figure 1 need to redraw give detailed mechanism stepwise.
- Is there any effect of temperature during optimization studies give details? If possible perform TEM analysis.
- Surprisingly very little discussion of results with the previous results of literature needs substantial discussion at the revision stage. Use recent references from the year 2018-2020.
- Techno Economic challenges of the developed Retinol-loaded Lipid Nanocarrier need to be addressed. Write the practical applications and future research perspectives and challenges by adding a new section before conclusions.
- What are the limitations to use this methodology for commercial application?
- The conclusion of the study is not discussed with the specific output obtained from the study, it could be modified with precise outcomes with a take home message.
- English and grammar mistakes are present. The author should check the manuscript by native English Speaker to improve the quality of the manuscript.
Reviewer 2 Report
Dear Authors,
The manuscript described a retinol nanostructure obtained by vacuum emulsification with characterisations and in vitro tests. Research was adequately justified and results were interesting, although there was not a deep discussion that must have been presented. Several paragraphs lacked references in Results and Discussion and that must be corrected.
- lines 62-63: more nanostructure-based strategies were used as DDS.
- lines 93-94: in our opinion, the system was not synthesised, but prepared. No synthesis was identified.
- emulsification method used high temperature. Was it compatible with the active (line 124)?
- please, present some data about the analytical method (chromatogram of active and excipients, for instance). please, reference the method.
- line 183: how many strips? Please, reference the method.
- is figure 1 an original figure?
- plase, clarify lines 231-233 and table 1. It was confusing if they were results from your research or referenced from another.
- line 276: what was the original color? a table summarising all characterisation of the samples would be very interesting.
- table 1 would be mor suitable in Material and Methods.
- line 302: please, explain flowability.
- table 4 was not self-explanatory.
- composition of commercial sample must be described, at least, qualitatively.
- line 461: it would be questionable that a conclusion present the term excellent properties. please, revise this adjective.
Round 2
Reviewer 1 Report
The authors have substantially revised the whole manuscript according to the comments.
The present form of the manuscript can b accepted for publication.
Reviewer 2 Report
Dear Authors,
Thank you for providing all responses.